# Using Photo Stories to Support Doctor-Patient Communication: Evaluating a Communicative Health Literacy Intervention for Older Adults

**DOI:** 10.3390/ijerph16193726

**Published:** 2019-10-03

**Authors:** Ruth Koops van ‘t Jagt, Shu Ling Tan, John Hoeks, Sophie Spoorenberg, Sijmen A. Reijneveld, Andrea F. de Winter, Sonia Lippke, Carel Jansen

**Affiliations:** 1Aletta Jacobs School of Public Health; 9747 AD Groningen, The Netherlands; 2Department of Communication and Information Sciences, University of Groningen, Oude Kijk in ’t Jatstraat 26, 9712 EK Groningen, The Netherlands; j.c.j.hoeks@rug.nl (J.H.); c.j.m.jansen@rug.nl (C.J.); 3Institute for Sport and Exercise Science, Department for Social Sciences of Sport, Westfälische Wilhelms-Universität Münster, Horstmarer Landweg 62a, 48149 Münster, Germany; shuling.tan@uni-muenster.de; 4Department of Health Sciences, University Medical Center Groningen, P.O. Box 196, 9700 AD Groningen, The Netherlands; s.l.w.spoorenberg@umcg.nl (S.S.); s.a.reijneveld@umcg.nl (S.A.R.); a.f.de.winter@umcg.nl (A.F.d.W.); 5Department of Psychology & Methods, Jacobs University Bremen, Res IV, Campus Ring 1, D-28759 Bremen, Germany; s.lippke@jacobs-university.de; 6Language Centre, Stellenbosch University, 7600 Stellenbosch, South Africa

**Keywords:** doctor-patient communication, health literacy, narratives, photo stories, older adults, self-efficacy, behavioral intention, format preference, effectiveness

## Abstract

Older adults often have limited health literacy and experience difficulties in communicating about their health. In view of the need for efficacious interventions, we compared a narrative photo story booklet regarding doctor-patient communication with a non-narrative but otherwise highly similar brochure. The photo story booklet included seven short picture-based stories about themes related to doctor-patient communication. The non-narrative brochure had comparable pictures and layout and dealt with the same themes, but it did not include any stories. We conducted two Randomized Controlled Trials (RCTs) among older adults with varying levels of health literacy: one RCT in Germany (*N* = 66) and one RCT in the Netherlands (*N* = 54); the latter one was followed by an in-depth interview study among a subset of the participants (81.5%; *n* = 44). In the RCTs, we did not find significant differences between the photo story booklet and the non-narrative brochure. In the interview study, a majority of the participants expressed a preference for the photo story booklet, which was perceived as recognizable, relevant, entertaining and engaging. We conclude that photo story booklets are a promising format but that there is room for improving their effectiveness.

## 1. Introduction

Low levels of health literacy (HL) have frequently been associated with poor health outcomes [1,2]. Adults with limited health literacy experience more difficulties in participating in care consultations, ask fewer questions and report less patient-centered communication [3]. Successful doctor-patient communication [4], for instance about shared decision making [5], critically depends on sufficient levels of communicative and general health literacy [6,7] and has been shown to be associated with patient satisfaction, clinical outcomes and adherence [8,9]. The negative consequences of low health literacy may be even more severe for older adults, who often have to deal with multiple chronic conditions and are more likely to suffer from cognitive and sensory declines such as hearing loss and memory problems [10]. For older adults, doctors and other health care providers appear to be the most trusted and the most frequently used source of health information, which makes appropriate doctor-patient communication even more relevant [11,12]. The importance of patient-provider communication for older adults with limited health literacy calls for the development and evaluation of interventions intended to educate, support and empower older patients in health care interactions [13]. According to recent studies, there is a need for simple interventions that use familiar language, address patient-provider communication and also address patients’ barriers such as insufficient self-efficacy [14,15,16]. 

Photo stories, also called ‘fotonovelas’, are small publications, often in booklet format, that tell a dramatic story by means of photographs and short and easily readable captions. Photo stories are increasingly used as health communication tools to educate, support and empower people with lower levels of health literacy with respect to various health subjects [17,18,19,20,21,22,23]. The integrated presentation of textual and visual information to support information processing and learning is in line with the principles of multimedia understanding, which posit that the integrated presentation of textual and visual information can minimize cognitive load and thus support learning [24,25,26].

Using photo stories as educational tools is an example of ‘entertainment education’, where messages are purposely designed as narratives that both entertain and educate, in order to increase knowledge and to change behavior. As Moyer-Gusé and Dale [27] explained in their overview of narrative persuasion theories, narratives are processed differently from other message formats. Other than readers of more traditional health messages, readers (or viewers) of a narrative can get involved in the story world (a process called ‘transportation’) that is created, and in the characters that play a role (‘identification’). An influential theory in the narrative persuasion literature, the Entertainment Overcoming Resistance Model (EORM) [28], posits that engagement with the narrative helps in overcoming resistance to behavioral changes, making narratives more effective than other persuasive messages. One of the antecedents of behavioral changes caused by reading or viewing a narrative, is the reduction of counterarguing against story-consistent beliefs. Similar to what is posed in the Extended Elaboration Likelihood Model (E-ELM) [29], the EORM states that transportation and identification will reduce the reader’s or viewer’s tendency to counterarguing. 

Finally, narratives are often built around main characters who have to overcome certain obstacles to reach their goals [30,31]. This makes the narrative format especially suitable for providing people with role models and strategies to overcome barriers in communicating effectively, for instance with their doctor, thereby increasing their levels of communication self-efficacy which is an important determinant of intentions and (health) behavior [32,33,34,35,36,37,38,39]. 

Based on these insights, we developed a narrative- and picture-based health literacy intervention, including seven very short photo stories on themes that older adults frequently mentioned during focus group discussions on doctor-patient communication (see [40] for details on development). The photo stories portrayed recognizable characters who were faced with a concrete communication problem that they successfully overcame. By implicitly providing the reader with step-by-step scenarios for solving similar problems, this intervention addressed patients’ barriers to successful communication and offered applicable strategies when patients encounter such barriers [41]. 

In view of the advantages of narrative health communication reported in the literature [27,29,31,42,43,44], we wanted to investigate whether the photo story booklet might outperform a non-narrative but otherwise highly similar brochure with respect to people’s beliefs that they could perform the communication strategies that the booklet and the brochure referred to (self-efficacy), and with respect to their intention to engage in those communicative behaviors (i.e., behavioral intention). In line with McGuire’s information-processing paradigm [45], paying attention to a message is a necessary first condition for any processing to take place. In other words, motivating the readers to start and continue processing the message is a prerequisite for effective health communication. For this reason, we also wanted to investigate possible preferences for the photo story booklet compared to its non-narrative competitor. 

Our first aim was, therefore, to assess the effects of a photo story booklet regarding doctor-patient communication on self-efficacy and behavioral intention, compared to a non-narrative, but otherwise highly similar brochure, among older adults with varying levels of health literacy. Second, we assessed whether older adults preferred the photo story booklet or the non-narrative brochure we used, and we investigated the reasons for possible differences in preferences. 

## 2. Materials and Methods 

### 2.1. Study Design

We concurrently conducted two Randomized Controlled Trials (RCTs). One RCT was conducted in Germany, comparing the German versions of the photo story booklet and a similar but non-narrative brochure among older participants with varying levels of health literacy. A second, similar RCT was conducted in the Netherlands, comparing the Dutch versions of the photo story booklet and the non-narrative brochure, also among older participants with different levels of health literacy. Due to organizational constraints in recruiting participants from the target groups, we could not include a traditional, ‘care as usual’ condition in the RCTs. Third, we conducted an in-depth interview study in the Netherlands, assessing the preference for either the photo story booklet or the non-narrative brochure. Unfortunately, we were not able to collect comparable data from our German participants.

The RCT in Germany was approved by the ethical board of the Deutsche Gesellschaft für Psychologie (DGPs; approval number SL062015) and is registered with ClinicalTrials.gov (number NCT02502292). The RCT in the Netherlands was approved by the Research Ethics Committee (CETO) of The Faculty of Arts, University of Groningen and is registered in the Netherlands Trial Register (number NTR5810). 

### 2.2. Participants

For the RCT in Germany, we recruited 66 participants with different levels of health literacy, aged between 54 and 94 years (30.3% male, *n =* 20), from senior day care and rehabilitation centers and sports clubs. The RCT in Germany was part of a larger RCT in which researchers also presented both interventions on a tablet [46]. In the present study, data collected in Germany were analyzed regarding the paper-and-pencil versions only. For the RCT in the Netherlands, we recruited 54 participants with lower and higher levels of health literacy, aged between 77 and 95 years old (34.5% male, *n* = 19), from an existing research database of participants of Embrace [47]. Embrace (Dutch: Samen Oud; Aging Together) is a population-based, person-centered and integrated care service for community-living older adults aged 75 years and older living in the Northern part of the Netherlands. To be concise, methods of both RCTs will be discussed together where possible. 

The interview study was conducted among all participants of the Dutch RCT. However, audio recordings could only be analyzed for 81.5% of the participants (*n* = 44). We could not include the remaining participants (*n* = 10) in our analysis because these participants did not give permission to record the interview or because technical problems arose during recording.

### 2.3. RCTs: Randomization

In both countries, we randomized the RCT participants to one of two conditions using numbers computed for each participant at entry of the study. Conditions regarded: 1) the photo story booklet or 2) the non-narrative brochure. Inclusion criteria were as follows: (1) age ≥ 50 years; (2) native speaker of German or Dutch, respectively; (3) pre-specified different levels of health literacy (see Section 2.8). For the flow of RCT participants in both countries, see Figure 1. 

### 2.4. Intervention: Photo Story Booklet

In our prior work, we developed a photo story booklet in multiple languages: German, Dutch, English, Italian and Hungarian. A participatory approach was followed, with contributions from older adults with limited health literacy. Co-creating the photo stories with members of the target group resulted in a booklet including seven one-page stories that the older adults involved considered relevant, recognizable and appealing. The positive feedback during a small-scale evaluation suggested that photo stories may indeed effectively support older adults with a low level of literacy; for detailed information, see [40]. After piloting a draft of the photo story booklet and the questionnaire to be used (see Section 2.5 and Section 2.6) among older participants in the Netherlands, we adapted the materials both in the Netherlands and in Germany to the groups with the lowest health literacy levels. The photo story booklet aims to increase older communicative self-efficacy and behavioral intention, as a means to support and empower patients during primary care consultations. The themes in the stories are: (1) general practitioner’s (GP) lack of attention; (2) bringing someone as support when stressed or nervous; (3) asking for plain language; (4) what to do when feeling uninformed about an overwhelming number of prescriptions; (5) implementation of lifestyle recommendations into concrete daily life actions; (6) medication management; (7) making a question prompt sheet before your consult. Each theme was incorporated into a brief photo story using photographs with realistic characters and vivid pictures and captions and text balloons (see Figure 2). 

### 2.5. Control Condition: Non-Narrative Brochure 

Based on the same content as the photo story booklet, we developed a non-narrative brochure. In this non-narrative brochure, the main messages of each of the seven stories in the photo story booklet were presented in the form of general advice. Each advice was accompanied by one large picture that was selected from the pictures that were taken for the photo stories (see Figure 3). The non-narrative brochure was developed as a ‘plausible rival’ to the photo story booklet, using a multiple-feature revision approach and considering evidence-based design principles for people with low health literacy (see [14]). The photo story booklet and the non-narrative brochure were designed using the same colors, paper, size and front page, and they portrayed the same characters.

### 2.6. Procedure

All German participants (*N* = 66) and all Dutch participants (*N* = 54) signed a consent form and agreed to participate. All 120 participants confirmed they were aware they could withdraw at any time, and approved of the fact that the results would be used and published for research purposes. The German participants (*N* = 66) were asked to fill out the demographic part of the questionnaire, including their perceived health, the frequency of their GP consultations, the AURA (Ask, Understand, Remember, Assessment, a brief measure of communication self-efficacy in clinical encounters; [48]), and the SBSQ (a Set of three Brief Screening Questions on health literacy; [49]) before they were instructed to read either the photo story booklet or the non-narrative brochure. After finishing reading, the German participants were instructed to fill out the second part of the questionnaire, which contained the items measuring self-efficacy and behavioral intention. In the Netherlands, after participants finished reading the photo story booklet or the non-narrative brochure, they were interviewed by research assistants, based on a pre-structured questionnaire that included both demographic questions as well as primary outcomes. The Dutch participants were not asked to fill out the SBSQ, as their SBSQ scores were already included in the Embrace database (see Section 2.2). Research assistants took note of the answers to the questionnaire. In both studies, participants were instructed to read at their own pace.

### 2.7. Measures

#### 2.7.1. Primary Outcomes in the RCTs

Self-efficacy and behavioral intention were measured for each of the communication themes addressed in the photo story booklet and the non-narrative brochure, with questions that could be answered on five-point Likert scales. For self-efficacy, questions were used such as: ‘Imagine you feel insecure about a visit to your doctor, or that you have the feeling that you could fall silent during the conversation. Do you think it would be easy for you to bring someone to support you?’ (1 = No, not at all; 5 = Yes, very much) (Germany: α = 0.65; the Netherlands: α = 0.81). Questions about behavioral intention included, for instance, ‘Imagine you feel insecure about a visit to your doctor, or that you have the feeling that you could fall silent during the conversation. Do you think you will bring someone to support you next time?’ (1 = No, certainly not; 5 = Yes, I certainly will) (Germany: α = 0.76; the Netherlands: α = 0.83). To enhance ease of processing, we added green checkmarks for positive answers, red crosses for negative answers and a question mark for ‘I don’t know’ to the answering scales of all items. Cronbach’s alphas were calculated for all scales (item scores were reversed where necessary). Mean values were only calculated when alpha was satisfactory (α ≥ 0.65).

#### 2.7.2. Preferences as Additional Outcomes in the In-depth Interview Study in the Netherlands

To assess possible preferences for the photo story booklet or the non-narrative brochure, all 54 Dutch participants, after having answered the questions for the RCT, were asked to compare the photo story booklet with the non-narrative brochure. Participants who had read the photo story booklet were asked to take a look at the non-narrative brochure now, and vice versa. Subsequently, participants were asked to indicate their preference and to provide an explanation. Answers were collected from 44 participants (a subset of 81.5% of the Dutch participants). We assessed preference for one of the two interventions by asking participants to indicate (Q1) which format they believed was the best and (Q2) which format they would like to take home. Participants were also asked to provide reasons for their preference.

### 2.8. Analyses

#### 2.8.1. RCTs

In both RCTs, for self-efficacy and behavioral intention a Repeated Measures Multivariate Analysis of Variance (MANOVA) was performed with the variable theme (theme1 to theme7; see Section 2.4) as a repeated measure, under which the variable measure (self-efficacy versus behavioral intention) was nested. Theme and measure were included in order to reduce variance related to possible differences between the separate themes included in both interventions and between both measures (i.e., self-efficacy and behavioral intention). Format (photo story booklet versus non-narrative brochure) and health literacy group (low versus high for the Netherlands; low, medium or high for Germany; see Section 3.1) acted as independent between-participant variables. For all analyses, we considered effects to be statistically significant at *p* < 0.05 (two-tailed). 

Power calculations using G*Power 3.1 (Heinrich Heine Universität Düsseldorf, Germany) ([50]) revealed that statistical power for finding large effects was 0.80 in the German RCT, and 0.72 in the Dutch RCT.

#### 2.8.2. In-depth Interview Study in the Netherlands

Preferences for either the photo story booklet or the non-narrative brochure expressed during the in-depth interviews and explanations for these preferences were analyzed both quantitatively and qualitatively. Verbatim transcripts of preference and explanations for preference were available for 44 interview participants. For the quantitative analysis, we counted the number of times the photo story booklet or non-narrative brochure was the preferred choice. The qualitative analysis was conducted by the first author (R.K.v.t.J.) using a data-driven approach based on the framework analysis method [51]. The analysis was guided by two questions: (a) “Which reasons do participants provide for preferring either the photo story booklet or the non-narrative brochure?” and (b) “How are these reasons related to basic principles of information processing such as attention and motivation, comprehension and action?” The categorization of the explanations provided was based on theoretical models of information processing (e.g., [45,52,53,54]), distinguishing the following communicative aspects: (1) attention for the message/motivation to process the message (attention and motivation), (2) ability to process the message (comprehension) and (3) subsequent mental and behavioral consequences of the message (action).

## 3. Results

### 3.1. RCTs: Participants

Table 1 provides a summary of participant characteristics of the RCTs in Germany and the Netherlands.

Between the samples from the two countries, significant differences were found for age (Dutch sample older), years of education (Dutch sample lower level of education), health (Dutch sample poorer health) and GP visiting frequency (Dutch sample reporting fewer GP visits). In neither of the two countries, however, there were significant differences between the two conditions (photo story booklet, non-narrative brochure) on any of the variables, except for age in the German sample. Here, mean age was significantly lower in the photo story booklet group compared to the non-narrative brochure group. 

Groups of participants differing in health literacy level were formed in both samples. In the German sample, the ‘low level’ group (*n =* 13) had health literacy sum scores below 6, the ‘medium level’ group (*n* = 12) had health literacy sum scores of 6 or 7, while the ‘high level’ group had health literacy sum scores of 8 and higher (*n* = 41). In the Dutch sample, the ‘low level’ group (*n* = 14) had health literacy sum scores below 6; the ‘high level’ group (*n* = 40) had health literacy sum scores of 8 and higher (corresponding to the cut-off point as used in [55]). Recruitment in the Netherlands was targeted specifically at the lower and higher health literacy groups in the Embrace database (see Section 2.2); in Germany, no such specific recruitment procedure was followed. As a result, participants with medium levels of health literacy were only present in the German sample.

### 3.2. RCT in Germany: Outcomes

No significant main effects or interaction effects of condition and health literacy level on self-efficacy or behavioral intention were found (*p* values > 0.20; η^2^ = 0.004−0.052). Table 2 shows the mean levels of self-efficacy and behavioral intention for both conditions and all health literacy groups. 

Excluding participants with medium levels of health literacy from these analyses did not essentially alter results.

### 3.3. RCT in the Netherlands: Outcomes

No significant main effects of condition on average levels of self-efficacy and behavioral intention were found (*F*(1,48) = 3.21; *p* = 0.079; η^2^ = 0.063). However, there was a significant main effect of health literacy level (*F*(1,48) = 11.01; *p* = 0.002; η^2^ = 0.187), with higher average levels of self-efficacy and behavioral intention for the higher health literacy group (4.43 (0.11) and 4.37 (0.11)), compared to the lower health literacy group (3.74 (0.18) and 3.76 (0.18)). Table 3 shows the levels of self-efficacy and behavioral intention for both conditions and both health literacy groups. No significant interaction effects of condition and health literacy group on self-efficacy and behavioral intention were found (*F*(1,48) = 0.57; *p* = 0.46; η^2^ = 0.012).

### 3.4. In-depth Interview Study in the Netherlands: Participants

Table 4 provides an overview of the characteristics of the 44 interview participants. A comparison between the RCT and the interview study in the Netherlands revealed no systematic differences in participant characteristics between the total group of Dutch participants and the subset of interview participants. 

### 3.5. In-depth Interview Study in the Netherlands: Outcomes

A statistically significant majority of participants (66.7%, z = 2.450, *p* = 0.014) preferred the photo story booklet over the non-narrative brochure (Q1). A statistically significant majority of participants (77.6%, *z* = 3.85, *p* = < 0.001) said they would rather take home the photo story booklet than the non-narrative brochure (Q2). Several participants (14.3%) indicated they would like to take home both the photo story booklet and the non-narrative brochure. Participants who had read the photo story booklet first did not significantly differ in preference from participants who had read the non-narrative brochure first: χ^2^(1) = 4.49, *p* = 0.11 for (Q1); χ^2^(1) = 5.07, *p* = 0.28 for (Q2). Notably, some participants reported they did not see themselves as belonging to the target group for the photo story booklet or the non-narrative brochure, but felt that they would be beneficial for acquaintances or relatives.

#### Explanations for Preference

Table 5 presents the most frequent types of explanations (with examples) for the preferences for either the photo story booklet or the non-narrative brochure as indicated during the interviews. The explanations suggest that the photo story booklet outperformed the non-narrative brochure when it comes to attracting attention (attention and motivation), processing ease (comprehension) and action (i.e., emotional and behavioral consequences) for most participants. Notably, factors related to Comprehension (comprehensibility) were mentioned most frequently to support people’s preference for both the photo story booklet and the non-narrative brochure. Photo stories were considered to be an appealing format, which could help people through the step-by-step examples. Participants who preferred the non-narrative brochure generally did so because they felt the general advice was shorter, simpler and more ‘to the point’, which mostly relates to comprehension. 

Both the photo story booklet and the non-narrative brochure seemed to encourage a small majority of participants (25 out of 44) to share their own experiences regarding doctor-patient conversations with the interviewers (action). 

Five participants referred to the photo story booklet in particular as supporting readers in making action plans for future scenarios: “Listen, in that one (photo story booklet) there are a lot more things that make you think: well I’m going to use that next time. For example, when you have some ailment, and they ask you—well I will make a list of my medicines.” (male, 79 years, higher HL). 

Several participants also noted that general advice may not be sufficient and that the specific real-life examples portrayed in the photo stories make the content more comprehensible: “This (non-narrative brochure) is not helpful for people who struggle. […] The pictures (referring to the photo story booklet) make it more comprehensible for people, because they can recognize themselves in the stories. […] For someone who’s afraid, this (non-narrative brochure) won’t help much, right?” (female, 85 years, higher HL; felt she did not belong to the target group). 

Finally, some participants explicitly suggested that the photo story booklet and the non-narrative brochure would work well together: “When you have this one (non-narrative brochure) and then read the other one (photo story booklet) after that, then you have all the details. When you read this one and you think ‘what does this mean?’ then you can read that in the other one. The first one is just the short version.” (female, 82 years, higher HL). 

## 4. Discussion

Our first aim concerned the effects of a photo story booklet regarding doctor-patient communication on self-efficacy and behavioral intention, compared to a non-narrative, but otherwise highly similar brochure, among older adults with low, medium and high levels of health literacy. In two RCTs performed in Germany and in the Netherlands, we found no significant differences between the photo story booklet and the non-narrative brochure for the group as a whole nor for any of the Health Literacy subgroups. Second, we assessed whether older adults preferred the photo story booklet or the non-narrative brochure we used, and we investigated the reasons for possible differences in preferences. In a follow-up interview study conducted among a subset of 81.5% of the Dutch participants, a statistically significant majority of the participants expressed a preference for the photo stories, which were perceived as recognizable, relevant, entertaining and engaging.

Similar outcomes are reported in a recent quantitative study in South Africa into the effects of a health-related photo story about crystal meth compared to a non-narrative brochure based on two existing traditional crystal meth documents [19]. In that study, the photo story on the dangers of crystal meth did not outperform the non-narrative brochure on knowledge or attitude. Only intention towards speaking to a family member or friend involved with crystal meth was significantly higher in the photo story condition. The photo story was clearly preferred, however, over the non-narrative brochure. This was especially the case for readers with low levels of education. 

The first explanation for the lack of significant differences in self-efficacy or behavioral intention that we found between the photo story booklet and non-narrative brochure in both RCTs may be the relatively small difference between the two formats. The non-narrative brochure was purposely designed as a ‘plausible rival’ of the photo story booklet, using a multiple-feature revision approach. It was carefully developed, considering evidence-based design principles for people with low health literacy (e.g., a multiple-feature revision approach; see [14]). In addition, the non-narrative brochure contained large photographs portraying the same characters that played a role in the photo story booklet. Perhaps these photographs have induced some form of narrative processing in readers of the non-narrative brochure (compare [56], where it was found that viewers of only one picture on a cigarette package with a short text line were able to conjure up a complete story).

Second, it should be noted that the measurements of self-efficacy and behavioral intention on the one hand and preference on the other one reflect different aspects of information processing. While no significant differences were found in effects related to action (self-efficacy and behavioral intention), the difference we did find in motivation (preference) suggests that in a natural context people might be more willing to read the photo story booklet than the non-narrative brochure. Evidently, willingness to read a given document is a necessary, though under-researched condition for any processing effect to occur (see also [19]).

Regarding preferences, a majority of participants in our third study distinctly preferred the narrative intervention because they found the photo story booklet recognizable, relevant, entertaining and engaging. Participants referred to possible positive effects of the photo story booklet by stating that the stories could help people through their step-by-step scenarios, that the photo stories could be read multiple times and still be interesting, and that they considered the photo story booklet to be an appealing format that would attract readers. 

Several participants felt that general advice as provided in the non-narrative brochure was not sufficient to support them in communicating with their doctor and that the specific, more elaborate examples portrayed in the photo stories helped to formulate action plans [36,57]. The photo stories seemed to help readers to bridge the intention-behavior gap [58]. The concrete behavioral and verbal responses embedded in the photo story scenarios provide ‘If-Then’ plans or implementation intentions, which have been shown to help people reaching goals [59]. This suggests that the photo story booklet not only informs and educates its readers, but also teaches them strategies, supports implementation intentions and thus behavioral change, which is in line with the core components of effective health literacy interventions as identified in, for instance, the Intervention Research on Health Literacy Among Ageing Population Project (IROHLA) (see [6,13,14,60]). Using visual narratives, such as photo stories, may be particularly effective in stimulating mental imagery, which is assumed to be an important factor in the formulation of implementation intentions [61,62]. While in both RCTs reading the photo story booklet or the non-narrative brochure proved to be associated with relatively high average levels of self-efficacy and behavioral intention, photo stories might help to turn those intentions into action plans, as is suggested by five participants’ comments in the interview study. 

A major strength of this study is the combination of studies in two different European countries using a mixed-methods design [63]. By conducting two RCTs and an interview study, we are able to provide valuable information on the effects of a photo story booklet and a non-narrative brochure on doctor-patient communication, and the preference for either format among a population of older adults with different levels of health literacy. By combining multiple methods, we gained insights on the outcomes of the health document interventions we tested as well as on the processes by which such interventions might achieve their effects. 

Some limitations of this study should be considered too. First, in both RCTs, the absence of a traditional, ‘care as usual’ condition made it impossible to detect the effectiveness of either the photo story booklet or the non-narrative brochure per se. Second, no data were collected on preferences from our German participants, thus, we cannot compare the two countries in that respect. Furthermore, while the German participants filled out the RCT questionnaires themselves, the answers of the Dutch participants were registered by research assistants. This may, to some extent, have resulted in different reactions. In addition, the two samples differed on a number of characteristics, such as age range, which may have added some random variation and thus have contributed to the null finding. Third, due to limited numbers of participants, this study was insufficiently powered to detect other than large effect sizes [50,64], while communicative interventions in the field of persuasion usually only result in small to medium effects [65]. Perhaps larger scaled experiments with the same materials that were used here might reveal effects that exist in reality but could not be detected in the present RCTs. In addition, some participants with lower levels of health literacy had difficulties answering the questionnaire, as was indicated by interviewers’ observations that they struggled with the hypothetical character of some statements [66]. We may thus have failed to detect more subtle effects of the photo stories.

## 5. Conclusions

While the RCTs revealed no statistically significant differences between the effects of the photo story booklet and the non-narrative brochure on self-efficacy and behavioral intention, participants in the interview study liked the photo story booklet more, felt more motivated, believed that they could understand the content easier and believed they could apply the information better to daily life in terms of action planning or implementation intentions. Perhaps it would be fruitful to combine standard formats of health communication (e.g., non-narrative advisory brochures for informing the target group) with narrative formats (e.g., photo stories, for motivating the target group). Future studies could establish to what extent such combined formats strengthen doctor-patient communication. 

New, larger scaled studies might again explore, especially for older adults with limited health literacy, the effects of photo story booklets on readers’ self-efficacy and behavioral intention when communicating with their doctor. To this end, it would be valuable to include pre- and posttest measurements, and to compare photo story booklets not only with non-narrative brochures such as used in this study but also with traditional, ‘care as usual’ formats of health communication. Furthermore, we would welcome studies that explicitly assess participants, preferably in semi-structured interviews, for long term effects on their behavior and the determinants thereof in real doctor-patient communication. Perhaps new studies could further elaborate on the idea of some participants mentioning that they themselves would not benefit from the photo novel or the non-narrative brochure, but that they knew other people who would.

In addition, our findings point out a need for studying the effects of health communication interventions on both attention, motivation and comprehension, as well as the impact of such interventions on knowledge, attitude, norms and behavior (action). Future studies could for instance explore which format is associated with higher rates of spontaneous pickups and reading behavior in natural contexts such as GP waiting rooms (attention and motivation), which format is easy to process and remember (comprehension) and whether the interventions may affect patients’ communication behavior in primary care consultations (action). Finally, it would be useful to explore whether combining the photo stories with the advice included in the non-narrative brochure would increase the effectiveness of this type of communicative health literacy intervention.

## Figures and Tables

**Figure 1 ijerph-16-03726-f001:**
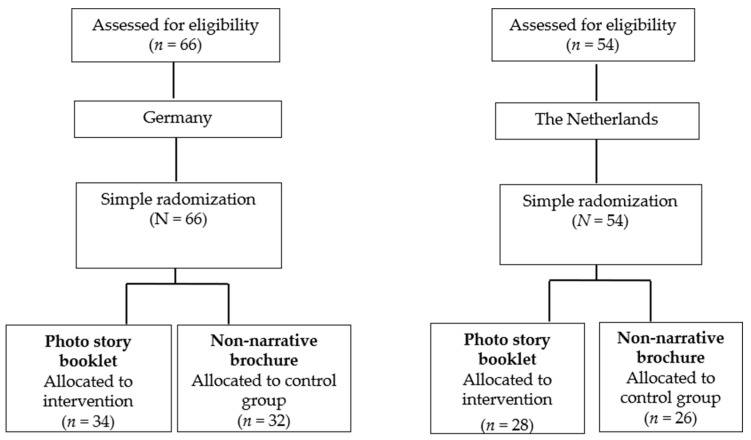
Flow of Randomized Controlled Trials (RCT) participants in Germany and the Netherlands.

**Figure 2 ijerph-16-03726-f002:**
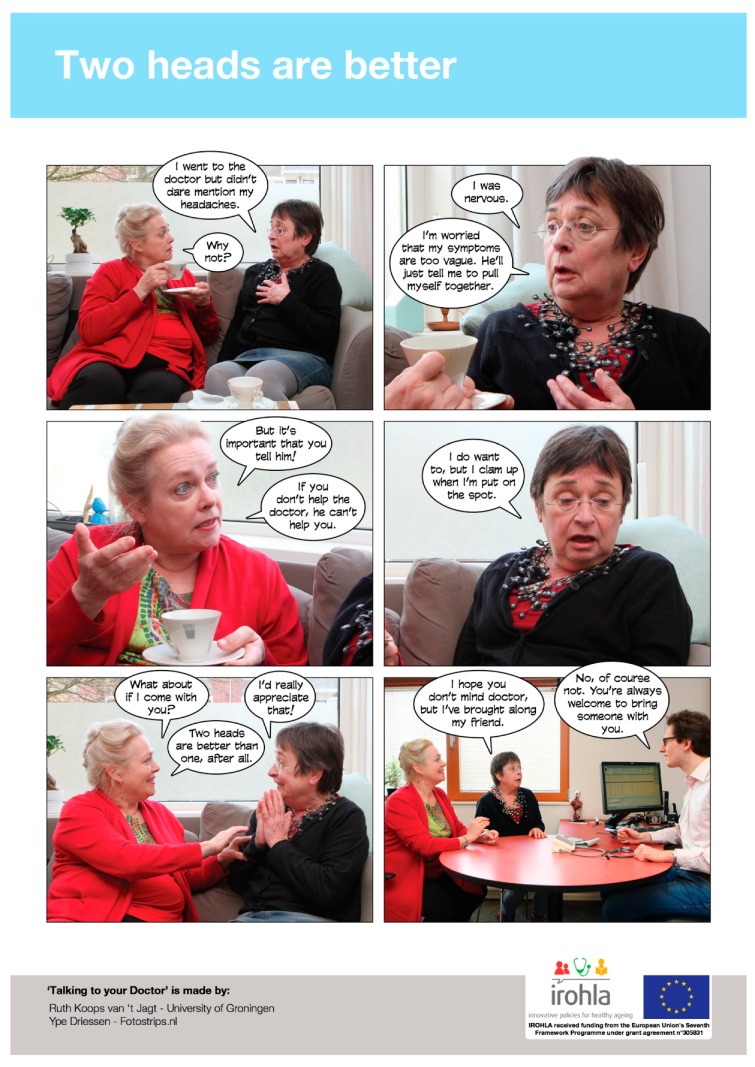
English version of the photo story about bringing someone for support.

**Figure 3 ijerph-16-03726-f003:**
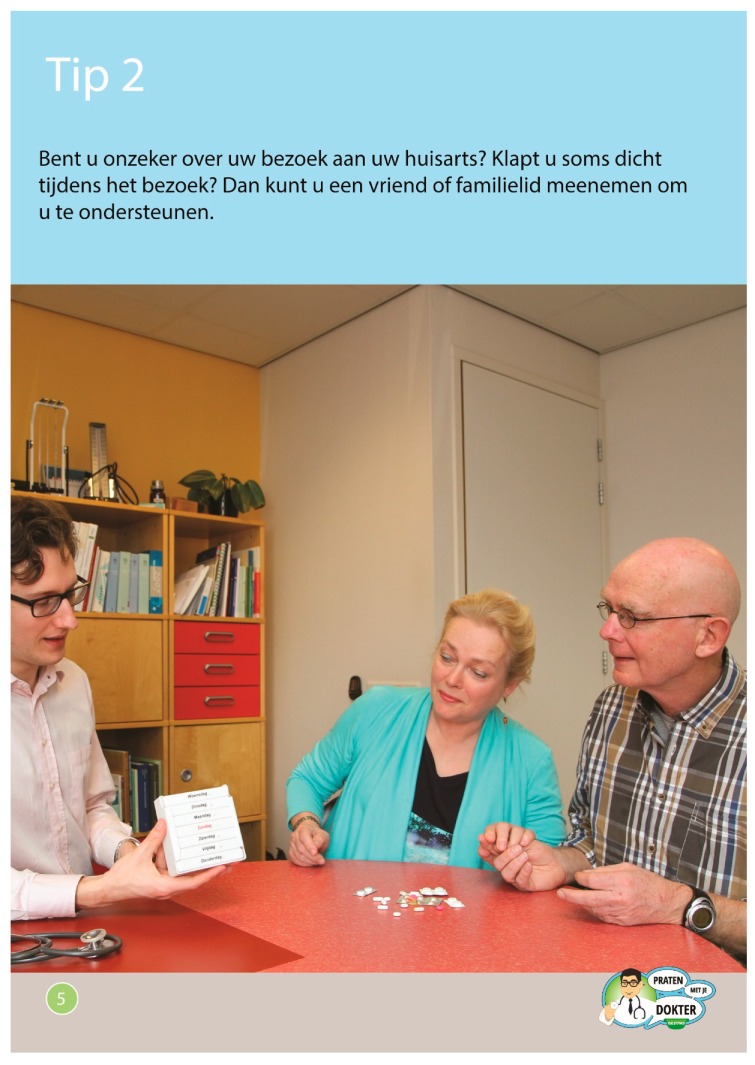
Example page in Dutch from the non-narrative brochure about bringing someone for support. Translation: Are you feeling insecure about visiting your General Practitioner? Are you sometimes at a loss for words during your visit? You can bring a friend or family member to support you.

**Table 1 ijerph-16-03726-t001:** Participant characteristics (means and standard deviations) in both RCTs.

	Germany			The Netherlands		
	Photo Story Booklet (*n* = 34)	Non-Narrative Brochure (*n* = 32)	Total (*N* = 66)	Photo Story Booklet (*n* = 28)	Non-Narrative Brochure (*n* = 26)	Total (*N* = 54)
Gender (female)	24	22	46	19	16	35
Age	73.2 (5.4)	76.8 (8.5)	75.0 (7.2)	82.1 (2.7)	83.6 (4.1)	82.8 (3.5)
Age (range)	62−80	54−94	54−94	77−88	79−95	77−95
Education (estimated in years)	10.74 (2.03)	10.72 (2.26)	10.73 (2.13)	8.96 (2.81)	8.69 (2.99)	8.83 (2.88)
Health (1 poor−5 excellent)	3.33 (0.85)	3.28 (0.96)	3.31 (0.90)	2.32 (0.86)	2.31 (0.62)	2.31 (0.75)
Visiting frequency general practitioner (GP) (1 less than once a year–6 at least once a week)	2.62 (0.99)	2.66 (0.97)	2.64 (0.97)	2.50 (1.07)	2.04 (1.04)	2.28 (1.07)
Health communication self-efficacy (AURA) (4 minimum−20 maximum)	15.76 (3.64)	16.22 (2.90)	15.98 (3.29)	15.79 (3.79)	17.35 (2.65)	16.54 (3.35)
Health Literacy (SBSQ) (0 minimum−12 maximum)	7.61 (2.28)	7.81 (2.24)	7.71 (2.24)	8.74 (2.98)	8.58 (3.08)	8.66 (3.00)

**Table 2 ijerph-16-03726-t002:** Average levels of self-efficacy and behavioral intention for both conditions for each Health Literacy Group (means and standard deviations) in Germany.

	Photo Story Booklet			Non-Narrative Brochure		
	Low HL	Medium HL	High HL	Low HL	Medium HL	High HL
Self-Efficacy	3.98 (0.68)	3.63 (1.01)	4.09 (0.69)	3.71 (0.88)	4.03 (0.71)	4.04 (0.63)
Intention	4.02 (0.78)	3.96 (0.87)	4.55 (0.62)	4.36 (0.37)	4.20 (0.72)	4.37 (0.53)

**Table 3 ijerph-16-03726-t003:** Average levels of self-efficacy and behavioral intention for both conditions for each Health Literacy Group (means and standard deviations) in the Netherlands.

	Photo Story Booklet		Non-Narrative Brochure	
	Low HL	High HL	Low HL	High HL
Self-Efficacy	3.65 (0.78)	4.22 (0.82)	3.82 (0.74)	4.71 (0.31)
Intention	3.63 (0.69)	4.23 (0.90)	3.88 (0.60)	4.59 (0.40)

**Table 4 ijerph-16-03726-t004:** Participant characteristics (means and standard deviations) in the interview study.

	Photo Story Booklet (*n =* 24)	Non-Narrative Brochure (*n* = 20)	Total (*N* = 44)
Gender (female)	16	13	29
Age	82.3 (2.7)	84.2 (4.5)	83.2 (3.7)
Age (range)	79−88	79−95	79−95
Education (estimated in years)	8.86 (2.90)	9.00 (3.23)	8.93 (3.02)
Health (1 poor−5 excellent)	2.38 (0.88)	2.25 (0.64)	2.32 (0.77)
Visiting frequency GP (1 less than once a year−6 at least once a week)	2.50 (1.10)	2.15 (1.09)	2.34 (1.10)
Health communication self-efficacy (AURA) (4 minimum−20 maximum)	15.92 (3.80)	16.95 (2.82)	16.39 (3.39)
Health Literacy (SBSQ) (0 minimum−12 maximum)	8.71 (2.90)	8.50 (3.46)	8.61 (3.13)

**Table 5 ijerph-16-03726-t005:** Explanations for preference grouped according to factors related to attention and motivation, comprehension and action.

	Photo Story Booklet	Non-Narrative Brochure
Number of Mentions	Illustrative Quotes	Number of Mentions	Illustrative Quotes
(Q1) Why does the respondent consider the photo story booklet or the non-narrative brochure to be the best?
Topic: Attention and motivation
Attractiveness	6	‘That one speaks to me more, it’s more pleasant.’ ‘I already know I like that one better, it’s more playful, I like it a lot more.’	1	‘This one is nicer of course… I think.’
Topic: Comprehension
Elaborateness	10	‘That one has more information.’ ‘In that one it’s explained a bit more.’	0	
Clearness	15	‘It’s very clear and the situations are explained very well.’ ‘The way it’s written makes everything very clear.’ ‘This is very clear and easy to understand, because of the stories.’	1	‘It’s clear.’
Comprehensibility	2	‘It’s written in a simple style.’ ‘This one reads quicker.’	4	‘Well that one’s a bit shorter.’ ‘If there’s one thing I hate it’s having to read a lot.’ ‘That one’s more to the point.’
Recognizability/relevance	4	‘In that one you see pictures. Like it is in daily life.’ ‘No, it’s all really familiar and recognizable for me.’	0	
Topic: Action
Mental processing	1	‘This one has more pictures, it’s much more visual.’	0	
Emotional appeal	3	‘Uhm, it gets to you more. It speaks to you.’	0	
Behavioral appeal	1	‘You respond to this one quicker.’	0	
(Q2) Why does the respondent want to take the photo story booklet or the non-narrative brochure home?
Topic: Attention and motivation
Attractiveness	4	‘Well that’s a pretty little book.’ ‘That one’s fun to read.’	0	
Topic: Comprehension
Elaborateness	11	‘Because there’s more to read in that one.’ ‘That one is just a little more elaborate.’	0	
Clearness	5	‘Clear example situations.’ ‘Clear information.’ ‘Easy to understand.’	0	
Comprehensibility	3	‘That one’s easier to read, I think.’	2	‘Because this one is simpler. They put it in a way I can really understand it.’
Recognizability/relevance	2	’And, well, those questions, they are very identifiable. When you’re at the doctors.’ ‘Because there are so many things in there that are recognizable to me.’	0	
Topic: Action
Mental processing	2	‘And if you look at those examples, it’s easier to remember.’ ‘That one makes you think more, I guess.’	0	
Emotional appeal	2	‘Because you, how do I explain this, can sympathize with that one more easily.’ ‘You really feel this one.’	0	
Behavioral appeal	1	‘You get more out of that one.’	0

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
