# Peer review of "Using Photo Stories to Support Doctor-Patient Communication: Evaluating a Communicative Health Literacy Intervention for Older Adults"

_ijerph, 2019, doi:10.3390/ijerph16193726_

Round 1

Reviewer 1 Report

This is an extensive research study which has the potential to contribute to the evidence base of public health and health care services literature. The focus on health literacy and older adult populations is a particularly important gap subject for investigation, and the authors have undertaken a considered approach to trialling suitable interventions to determine correlations with the extent of self-efficacy and behavioral intent. 

The paper can be strengthened, however, using additional clarifications.  The use of the term 'RCT' (randomized controlled trial) needs to be explained early in the paper.  Otherwise the use of the term is somewhat misleading.  For example, there is only a passing mention of how participants across the geographical sites were randomized.

Proper randomisation reduces selection bias at trial entry and is a crucial component of RCTs-  how has this been achieved - statistically and descriptively? 

Set out clearly any feasibility or pilot study preceding this full-scaled RCT.  There is mention of an earlier RCT trial - i.e. the trial conducted in Germany.  Is this replicable and how was it applied in similar measure across the other trials. 

Generally, the participant sizes might be seen as relatively small for effective randomisation and null hypothesis.   Statistical comparisons are likely to be misleading due to the potential problems arising from small samples and low power.  

In the current project context, the participant recruitment and selection process appear not to be truly randomised and the statistical analyses do not align with a full-scale RCT due to the variability and limitations in trial implementation at each of the sites.  The most common confusions are the characterization of a too-small RCT as a pilot study, and/or reporting treatment effectiveness results from a RCT in the same way that those from a pilot might be reported.

The larger German study is categorised, for example, as an Interventional Clinical Trial within the ClinicalTrials.gov database, however it is not clear if the present study fulfills the same criteria as it is presented. 

There is inconsistent methodological information across the study geographies to determine whether any of these fall within a RCT process, so it is critical that the authors provide adequate description of the overall study to closely match up with the chosen methods in use for this overall study. 

---

Other considerations:

The title is composed of two statements - is one a supporting subtitle?  If so, use a colon - as follows: " Using photo stories to support doctor-patient communication: evaluating a communicative health literacy intervention for older adults" .

RCT and other acronyms in the paper- make sure these are shown as expanded terms in the first text instance. RCT is not clarified, for example.

The Health Literacy Questionnaire (HLQ) is a common instrument in assessing HL sums combined with Rasch analysis.   It might be useful for the reader to include a justification/reasons for implementing the SBSQ measure in this study and not HLQ.  

Limitations - A considered section of limitations and challenges.  The absence of a pre-test or a ‘care as usual’ condition is an important limitation of the proposed interventions which should be brought up earlier in the study. Also the fact that qualitative data was not collected in the German trial -this is not adequately addressed in the methodology or discussion.  This has possible implications on the 'RCT' parameters.

Conclusion - this section could be expanded, perhaps touching on how to overcome some of the limitations in future work (e.g. limitations raised in discussion).  Insightful observations should also be included for the consideration of readers who may wish to build on this work.

Reviewer 2 Report

Dear authors,

Novel and nicely written manuscript!

I have a few comments:

introduction: some of your references are very old, for example reference 7 and 8. There are more up to date references about doctor patient communication and also about elderly patients and patients with limited health literacy. See for example publications in the journal Patient Education and Counseling. Materials and methods; could you explain 'the multiple feature revision approach'. How was the brochure developed? This will provide more insight into the comparability with the photo booklet. Is it possible to use sub-headings in the method and materials section? This would help. You mention that the German participants filled in the questionnaire and that the participants in the Netherlands were interviewed based o a pre-structured questionnaire. The interview method seems more appropriate for people with low/limited health literacy. Did the German participants comprehend the questions? I think you should mention this in your limitation section. You mention (in 3.4.2 Preference) that some participants did not see themselves as belonging to the target group for both booklets. Why not? And did you ask what they prefer/would be helpful for them? Perhaps you could add this to your discussion section.

Please check your manuscript for minor spelling errors.

Good luck!

Round 2

Reviewer 1 Report

Thank you to the authors for their significant work in outlining in fuller detail the RCT methodologies deployed in the field studies.   The resulting edited material provides a very solid foundation for understanding the research processes and for their replicability in future research.  A much stronger and considered conclusion, and overall, a strong contribution to the field of health literacy research.

------

Just a few minor suggestions for further clarity:

Line 196 -97.  "Figure 3. Example page in Dutch from the non-narrative brochure about bringing someone for support [.] Translation: Are you feeling insecure about visiting your GP? Bring a friend or family member to support you."

-Sight edit - addition of full-stop.

Line 131 -"Reasons for not including the remaining participants (n=10) were the refusal of some participants to record the interview, and technical issues."

-Slight edit for clarity. 

Line 167 - "In our prior work, we developed a photo story booklet in multiple languages."  

-  Suggest to qualify by naming a few of the languages as examples

Lines 200-201  "All participants signed a consent form and agreed to participate. All participants confirmed they were aware they could withdraw any time, and approved of the fact that the results would be used and published for research purposes."

-Suggest to qualify number of 'All' participants (n=X) in each of the two sentences.  Do you mean 'All' as a combination of the German and Netherlands trials?  Perhaps qualify this. 

Line 202 "In Germany, participants ...  "   Optionally give (n=x) to reiterate number of participants and to make distinct from 'All' participant numbers.   

Line 242-244 "In both RCTs, for self-efficacy and behavioral intention a repeated measures MANOVA was performed with a specified theme (themes 1 to theme7 - see section 2.4) as a repeated measure, under which measure (self-efficacy versus behavioral intention) was nested." 

-  Slight edit and suggestion to refer to the section (2.4) where the themes were listed.
